# The Impact of the COVID-19 Pandemic on Dietary Patterns of Pregnant Women: A Comparison between Two Mother-Child Cohorts in Sicily, Italy

**DOI:** 10.3390/nu14163380

**Published:** 2022-08-17

**Authors:** Roberta Magnano San Lio, Martina Barchitta, Andrea Maugeri, Maria Clara La Rosa, Giuliana Giunta, Marco Panella, Antonio Cianci, Fabiola Galvani, Elisa Pappalardo, Giuseppe Ettore, Antonella Agodi

**Affiliations:** 1Department of Medical and Surgical Sciences and Advanced Technologies “GF Ingrassia”, University of Catania, 95123 Catania, Italy; 2Obstetrics and Gynecology Unit, Department of General Surgery and Medical Surgical Specialties, University of Catania, via S. Sofia, 78, 95123 Catania, Italy; 3Obstetrics and Gynecology Unit, Azienda di Rilievo Nazionale e di Alta Specializzazione (ARNAS) Garibaldi Nesima, 95125 Catania, Italy

**Keywords:** SARS-CoV-2, COVID-19, diet, nutrition, pregnancy

## Abstract

A maternal diet, before and during pregnancy, plays a key role in ensuring maternal and newborn health. The COVID-19 pandemic, however, may have compromised dietary habits in the general population and in specific subgroups of individuals. Here, we evaluated the impact of COVID-19 on the diet of pregnant women, using data from two mother-child cohorts in Sicily (Italy). Dietary data were collected using a food frequency questionnaire and analyzed through the Mediterranean diet (MD) score and principal component analysis (PCA). The comparison of maternal dietary consumption before and during the COVID-19 pandemic showed differences in terms of vegetables (*p* < 0.001), fruit (*p* < 0.001), dairy products (*p* < 0.001), fish (*p* < 0.001), and legumes (*p* = 0.001). Accordingly, after adjusting for covariates, mothers enrolled during the pandemic were more likely to report low adherence to MD than those enrolled before (OR = 1.65; 95%CI = 1.12–2.42; *p* = 0.011). A similar result was obtained by analyzing the adherence to a prudent dietary pattern, derived through PCA and characterized by high intake of cooked and row vegetables, legumes, fruit, fish, and soup. Overall, these findings suggested that the COVID-19 pandemic may have influenced maternal diet during pregnancy. However, further efforts are needed to investigate the main causes and consequences of this change.

## 1. Introduction

Several COVID-19 containment measures (e.g., smart working and the limitation of outdoor activities), enforced to counter the spread of SARS-CoV-2, have entailed changes in lifestyle. For instance, increased stress may have affected individual eating habits, resulting in higher intake of alcohol, overeating of “comfort foods” rich in sugar, and energy imbalance [1]. Moreover, limited access to daily shopping may have reduced fresh food consumption (e.g., fruit and vegetables) in favor of highly processed products (e.g., junk foods and snacks) [2,3]. Several studies reported both positive and negative changes in dietary habits, as well as in the adherence to specific dietary patterns, of the general population due to COVID-19 lockdown [2,3,4,5]. The Mediterranean diet (MD) is commonly accepted as an ideal dietary model for the prevention and control of obesity and non-communicable diseases during one’s lifetime [6,7,8,9,10]. In particular, the reduced availability of material resources due to the COVID-19 pandemic has caused a general reduction in the adherence to MD [11]. In this scenario, pregnant women were considered at higher risk for adverse lifestyle changes in response to the COVID-19 pandemic due to multiple sources of perceived stress [12]. During pregnancy, the maternal diet plays a key role in ensuring fetal growth, with short- and long-term effects on maternal and child health [13,14,15,16,17]. Thus, pregnancy is considered as a period of nutritional vulnerability, in which nutrient requirements are increased and a well-balanced diet should be guaranteed. However, social restriction may have compromised women nutritional choices and dietary habits [18]. During the pandemic, a higher proportion of pregnant women reported dietary imbalance if compared with the pre-pandemic period. [19]. However, findings on this public health issue are still controversial [20] or not always evident [21]. In fact, there is currently a lack of evidence about the impact of the COVID-19 pandemic on the dietary habits of pregnant women. Studies have usually focused on the impact of COVID-19 on maternal anxiety, depression, and post-traumatic stress disorder [22,23,24]. Moreover, important determinants for pregnancy health—including the consumption of healthy foods—became less accessible during the pandemic, thus also acting on physical and mental health [25]. Although the promotion of maternal dietary recommendations during the COVID-19, the pandemic is expected to have dramatic indirect effects, with increasing risks of obesity and other diet-related diseases later in life [26,27]. Given the need to better understand the impact of COVID-19 on the maternal diet, the present study aims to assess dietary habits of pregnant women before and during the COVID-19 pandemic, using data from two birth cohorts in Catania, Italy.

## 2. Materials and Methods

### 2.1. Study Design

This is a cross-sectional analysis of data obtained from two prospective cohorts: the “Mamma and Bambino” and the “MAMI-MED” cohorts. Each cohort recruits mother–child pairs to evaluate how the exposome affects the health of the mothers and children. The “Mamma and Bambino” cohort was established in November 2014 and recruited more than 400 mother–child pairs until December 2019. The recruitment takes place during the prenatal genetic counselling at the Azienda Ospedaliero Universitaria Policlinico “G. Rodolico-San Marco” (Catania, Italy) and full details and protocols have been previously published [13,14,28,29]. The “MAMI-MED” cohort was established in December 2020, after the beginning of the COVID-19 pandemic. This ongoing cohort adopts a study design similar to the “Mamma and Bambino” cohort, but the recruitment takes place during the first trimester visit at the Azienda di Rilievo Nazionale e di Alta Specializzazione (ARNAS) Garibaldi Nesima (Catania, Italy). Both cohorts share the same study protocols, which are in line with the Declaration of Helsinki. For the “Mamma and Bambino” cohort, the study protocol was approved by the ethics committee of the “Azienda Ospedaliero-Universitaria Policlinico-Vittorio Emanuele” and by the ethics committee “Catania 1” with the following protocol numbers: 47/2014/VE; 48/2015/EMPO; 186/2015/EMPO; 197/2016/EMPO; 213/2017/EMPO; 231/2018/EMPO; 263/2019/EMPO). For the “MAMI-MED” cohort, the study protocol was approved by the ethics committee “Catania 2”, with the protocol number 487/CE, 71/2020/CECT2. All women are fully informed of the purpose and procedures and give their written informed consent to participate. For the purpose of the current cross-sectional analysis, we compared maternal dietary data obtained from the “Mamma and Bambino” cohort before the COVID-19 pandemic, with those obtained from the “MAMI-MED” cohort during the pandemic (i.e., from December 2020 to January 2022). The analysis was conducted on mothers with available socio-demographic and dietary information and an outline of the methodology is depicted in Figure 1.

### 2.2. Data Collection

In both cohorts, socio-demographics and lifestyles information was assessed at enrollment, by trained epidemiologists through structured questionnaires [13,14,28,29,30,31,32]. Educational level was categorized as low (primary education), medium (secondary education), or high (tertiary education). According to employment status, women were categorized into employed (including full-time or part-time employment) or unemployed (including retired). Regarding smoking status, women were categorized as smokers or non-smokers (including former smokers). BMI was calculated by measuring body weight and height and reported as kg/m2. According to the World Health Organization criteria [33], pregnant women were then classified as underweight, normal weight, overweight, or obese.

### 2.3. Dietary Assessment

In both cohorts, maternal diet was assessed at recruitment, using a food frequency questionnaire (FFQ) referring to the previous month [34,35]. This FFQ was adapted from a previous instrument validated for assessing folate intake of Italian women [36]. During the first interview, pregnant women were asked to indicate the frequency of consumption and serving size for a set of 95 foods and beverages, even with the support of an indicative photograph atlas. For each item, dietary intake was obtained as the product of frequency of consumption and portion size, and adjusted for total energy intake by the residual method [37]. Next, the Mediterranean diet score (MDS) was calculated to assess the adherence to Mediterranean diet (MD). Its computation was based on the ideal/poor consumption of nine food categories: fruits and nuts, vegetables, legumes, cereals, fish, dairy and meat products, alcohol, and the unsaturated/saturated lipid ratio [38]. In the current study, however, no women consumed alcohol and hence the MDS ranged from 0 to 8. For this reason, pregnant women were categorized as low-adherents (MDS from 0 to 3), medium-adherents (MDS from 4 to 6), or high-adherents (MDS from 7 to 8).

### 2.4. Principal Component Analysis

As described elsewhere, the principal component analysis (PCA) was used to derive a posteriori dietary patterns from a multivariate dietary dataset [35]. Before applying PCA, however, the 95 food items assessed by the FFQ were grouped into 39 predefined food categories, in accordance with their nutrient profiles and culinary usage. Next, PCA with varimax rotation was applied on energy-adjusted values to obtain a set of principal components reflecting different dietary patterns. To determine the number of informative dietary patterns, we used the following criteria: scree plot examination, eigenvalues > 2.0, and interpretability. For each dietary pattern, the factor score was obtained as the sum of products between energy-adjusted intakes and factor loadings. According to factor scores, pregnant women were categorized as low-adherents (first tertile of factor score), medium-adherents (second tertile), or high-adherents (third tertile). Food categories characterizing each dietary pattern were defined as those with factor loading ≥|0.4|.

### 2.5. Statistical Analysis

Statistical analyses were performed using SPSS v.25. Frequencies (percentage, %) or median and interquartile range (IQR) were used for descriptive statistics due to the skewness of quantitative variables. Bivariate analyses were conducted using the Mann–Whitney test for quantitative variables and the Chi-squared test for categorical variables. We also applied linear and logistic regression models to test the impact of COVID-19 pandemic on maternal dietary habits. Specifically, we performed a linear regression model using MDS as dependent variable and the following predictors: source of mothers (e.g., “Mamma and Bambino” or “MAMI-MED” cohorts), maternal age, gestational age at recruitment, educational level, employment status, pre-pregnancy BMI, and smoking status. We also applied separate logistic regression models using each food category of MD or the adherence to MD as the dependent variable and the above-mentioned predictors. With respect to the adherence to dietary patterns, we performed unadjusted and adjusted logistic regression models, using low vs. high adherence as dependent variable. All statistical tests were two-sided and *p*-values < 0.05 were considered statistically significant.

## 3. Results

### 3.1. Characteristics of Study Population

This cross-sectional analysis included 1198 pregnant women, 397 of whom were recruited in the “Mamma and Bambino” cohort before the beginning of the COVID-19 pandemic (median age = 37 years) and 801 of whom enrolled in the “MAMI-MED” cohort during the COVID-19 pandemic (median age = 31 years). Specifically, women were recruited in the “Mamma and Bambino” and “MAMI-MED” cohorts at a median gestational age of 16 and 12 weeks, respectively. As shown in Table 1, the comparison of maternal characteristics before and after the beginning of COVID-19 pandemic suggested differences in terms of age (*p* < 0.001), gestational age at recruitment (*p* < 0.001), pre-pregnancy BMI (*p* = 0.049), educational level (*p* = 0.002), and employment status (*p* = 0.005).

### 3.2. The Impact of COVID-19 Pandemic on Maternal Consumption of Foods

We first analyzed the ideal/poor consumption of main food categories, such as fruits and nuts, vegetables, legumes, cereals, fish, dairy products, meat products, and the ratio of unsaturated to saturated lipids. In particular, as shown in Table 2, the comparison of maternal dietary consumption before and during COVID-19 pandemic revealed differences in terms of vegetables (*p* < 0.001), fruit (*p* < 0.001), dairy products (*p* < 0.001), fish (*p* < 0.001), and legumes (*p* = 0.001).

Accordingly, women enrolled during the pandemic were more likely to report poor consumption of vegetables (OR = 2.12; 95%CI = 1.66–2.72; *p* < 0.001), fruit (OR = 2.90; 95%CI = 2.25–3.73; *p* < 0.001), legumes (OR = 1.53; 95%CI = 1.20–1.95; *p* = 0.001), fish (OR = 1.68; 95%CI = 1.32–2.14; *p* < 0.001), and dairy products (OR = 1.71; 95%CI = 1.34–2.18; *p* < 0.001). Since there were differences between women enrolled before and during the COVID-19 pandemic, we adjusted the analyses for potential confounders. In particular, women enrolled during the pandemic were more likely to report poor consumption of vegetables (OR = 1.85; 95%CI = 1.13–2.70; *p* = 0.001), fruit (OR = 3.30; 95%CI = 2.26–4.82; *p* < 0.001), legumes (OR = 2.13; 95%CI = 1.47–3.09; *p* < 0.001), and dairy products (OR = 1.47; 95%CI = 1.04–2.10; *p* = 0.03), after adjusting for age, gestational age at recruitment, educational level, employment status, pre-pregnancy BMI, and smoking status. Fish was the only food category that did not show association in the adjusted regression model (OR = 1.32; 95%CI = 0.89–1.78; *p* = 0.063).

### 3.3. The Impact of COVID-19 Pandemic on Maternal Adherence to Mediterranean Diet

In line with previous findings, we tested differences in the adherence to MD between women enrolled before and during the COVID-19 pandemic. Interestingly, the MDS was higher among women enrolled before COVID-19 pandemic (median = 4.3; IQR = 2.0) than in those enrolled during the pandemic (median = 3.9; IQR = 2.0; *p* < 0.001). In line, pregnant women enrolled during COVID-19 pandemic exhibited a ~0.4-point reduced MDS compared with mothers enrolled before the pandemic (β= −0.386; SE = 0.132; *p* = 0.003), after adjusting for age, gestational age at recruitment, educational level, employment status, pre-pregnancy BMI, and smoking status. Similarly, the proportion of medium or high adherence to MD was higher among women enrolled before than those enrolled during the pandemic (61.7% vs. 54.1% for medium adherence; 8.8% vs. 3.2% for high adherence; *p* < 0.001). In fact, mothers enrolled during the pandemic were more likely to report low adherence to MD (OR = 1.78 95%CI = 1.38–2.31; *p* < 0.001). This result was confirmed by logistic regression model that adjusted for age, gestational age at recruitment, educational level, employment status, pre-pregnancy BMI, and smoking status (OR = 1.65; 95% CI = 1.12–2.42; *p* = 0.011).

### 3.4. The Impact of COVID-19 Pandemic on Maternal Dietary Patterns

In the whole dataset, we next derived two major dietary patterns with eigenvalues ≥ 2.0, which explained 15.6% of the total variance among 39 predefined food groups. Figure 2 shows factor loadings, which explain the correlation between each food group and dietary pattern. The first dietary pattern—named “prudent”—was characterized by high intake of cooked and row vegetables, legumes, fruit, fish, and soup. By contrast, the second dietary pattern—named “western”—was characterized by high intake of white bread, vegetable oil, fries, salty snacks, dipping sauces, and sweets. For each dietary pattern, we classified maternal adherence according to tertile distribution of factor scores (i.e., low adherence for 1st tertile vs. high adherence for the 3rd tertile). Thus, we performed logistic regression models to evaluate the impact of COVID-19 on the adherence to dietary patterns. In the unadjusted model, mothers enrolled during the COVID-19 pandemic were less likely to adhere to the prudent dietary pattern than those enrolled before (OR = 0.22; 95% CI = 0.16–0.30; *p* < 0.001). By contrast, they were more likely to adhere to the western dietary pattern than their counterpart (OR = 1.60; 95% CI = 1.18–2.17; *p* = 0.002). However, the association remained significant only for the prudent dietary pattern, after adjusting for age, gestational age at recruitment, educational level, employment status, pre-pregnancy BMI, and smoking status (OR = 0.26; 95% CI = 0.15–0.43; *p* < 0.001). No significant association was evident for the adherence to the western dietary pattern (OR = 1.41; 95% CI = 0.88–2.01; *p* = 0.086).

## 4. Discussion

To the best of our knowledge, there is a lack of evidence about the relationship between COVID-19 pandemic and nutritional choices of pregnant women, one of most vulnerable groups that have faced challenges to their antenatal care, medication use, and childbirth [39]. To overcome this gap, our study aimed to compare dietary habits—in terms of adherence to MD and to specific dietary patterns—of pregnant women before and during the COVID-19 pandemic, using data from two Sicilian prospective cohorts. During pregnancy, the pandemic has been reported to affect lifestyle, including maternal diet. Although several studies have demonstrated that pregnant women experienced dietary changes during the COVID-19 pandemic [20,40], others did not find differences in nutritional choices before and during the COVID-19 pandemic [21]. Thus, evidence is still controversial, probably depending on study population and/or different timing of observation. In fact, the majority of studies were conducted during the COVID-19 lockdown, whereas others immediately after or with a longer time period from lockdown. In this scenario, our analyses were conducted comparing data of pregnant women enrolled before (from November 2014 to December 2019) and during the COVID-19 pandemic (from December 2020 to January 2022).

Firstly, we observed differences in the ideal/poor consumption of main food categories—typical of MD—before and during the COVID-19 pandemic. Our results were confirmed by the adjusted logistic regression model, showing that women enrolled during the pandemic were more likely to report poor consumption of vegetables, fruit, legumes, and dairy products. Similarly, during the most severe period of COVID-19 restrictions in 2020, Chen and colleagues assessed variations in food intake among pregnant women, which reported less consumption of vegetables, fruit, dairy products and nuts [19]. In general, there has been a dramatic impact on food production chain, including livestock breeding, processing, distribution and storage [41]. These findings supported the hypothesis that these differences could depend on the disrupted access to food during COVID-19 pandemic, as well as on food shortages and increased prices [42,43]. As reported by previous studies, the availability of some food groups resulted, for instance, in the reduced consumption of fruit and vegetables [44]. Moreover, the pandemic had a negative impact on employment and income, leading to a decrease consumption of healthy but expensive food, such as vegetables and dairy products [45].

Since MD is traditionally characterized by higher consumption of vegetables, fruit, legumes, whole grains and fish, moderate consumption of dairy products and alcohol, and lower consumption of meat, we tested differences in the adherence to MD between women enrolled before and during the COVID-19 pandemic. Interestingly, women enrolled before COVID-19 were more likely to adhere to MD. In particular, nearly 62% and 54% of women enrolled before and during the pandemic moderately adhered to MD, respectively. Similarly, we observed that the proportion of women that highly adhered to MD was higher among those enrolled before COVID-19. In this context, a previous analysis of the impact of confinement on pregnant women lifestyle showed that their adherence to MD was lower than desirable, with approximately two-thirds of women with poorer diet than recommendations [21]. It is worth noting that high adherence to the MD is associated with beneficial effects during pregnancy—in terms of preventing gestational diabetes, excessive gestational weight gain, and obesity [46]. However, COVID-19 might affect the adherence to MD, exacerbating other pre-existing conditions, such as obesity and metabolic syndrome [47,48,49]. In our study, pregnant women enrolled during the COVID-19 pandemic exhibited a ~0.4-point reduced MDS compared with mothers enrolled before the pandemic, after adjusting for potential confounders. In line, our findings were also supported by the fact that mothers enrolled during the pandemic were more likely to report low adherence to MD than those enrolled before. However, since evidence is scarce and conflicting, it is not yet clear whether and how COVID-19 pandemic influenced the adherence to the MD. In the general population, a systematic review demonstrated that the adherence to MD might have increased in some settings, while the determinants involved should be further explored [11]. In Italy, for instance, Di Renzo and colleagues demonstrated that population group aged 18–30 years resulted in having a higher adherence to MD when compared to the younger and the elderly population [3]. Thus, there is a need to investigate the impact and the long-term consequences of COVID-19 on dietary habits, while also considering the major determinants involved in this relationship.

In our study, we assessed the impact of COVID-19 pandemic on maternal dietary patterns. Thus, we derived two dietary patterns characterizing the whole dataset, which were analogous to those obtained by previous analyses on the “Mamma and Bambino” cohort [14,29]. This was expected since PCA is one the of the most common data-driven techniques to derive a posteriori dietary patterns from a multivariate dataset. For consistency with our previous works, we named “prudent” the dietary pattern characterized by high intake of cooked and row vegetables, legumes, fruit, fish, and soup, while the “western” dietary pattern was characterized by high intake of white bread, vegetable oil, fries, salty snacks, dipping sauces and sweets. It is worth underlining that slight divergences from previous dietary patterns depended on some differences in the study populations. Interestingly, we demonstrated that mothers enrolled during the COVID-19 pandemic were less likely to adhere to the prudent dietary pattern, and more likely to adhere to the western one. However, the association remained significant only for the prudent dietary pattern, after adjusting for age, gestational age at recruitment, educational level, employment status, pre-pregnancy BMI, and smoking status. As reported previously, dietary habits have changed in Poland, Austria, and the United Kingdom, as a result of the pandemic scenario, and with severe consequences on obesity and health-related problems [50]. In this context, a review conducted to assess dietary changes during the first lockdown stated that the impact of COVID-19 lockdown both negatively and positively affected dietary practices throughout Europe and globally. In particular, negative dietary habits were associated with other adverse outcomes, including weight gain and mental health [2]. This scenario also regarded pregnant women, which both improved and worsened their dietary habits. Particularly, negative changes in behaviors during the COVID-19 pandemic were related to pregnancy complications and socio-economic disadvantages [20]. The lack of any solid evidence in this field of research, therefore, encourages further efforts to understand the influence of COVID-19 pandemic on maternal dietary habits. Furthermore, future studies investigating maternal factors involved in this relationship could help to better understand the long-term effects of COVID-19 pandemic, and to translate the answers into effective public health strategies.

Our work had also some limitations that should be considered when interpreting our results. Firstly, our study compared maternal dietary habits obtained through cross-sectional analyses of two different cohorts, established before and during the COVID-19 pandemic. For this reason, our results should be interpreted cautiously and confirmed by further prospective studies. Secondly, women included in the two cohorts exhibited some differences that could affect our findings. Although we adjusted the analyses for potential confounders, the presence of residual confounders cannot be completely ruled out. Secondly, the limited sample size and the lack of sufficient data on birth outcomes did not allow us to perform additional analyses. Thus, the objective for the future is to evaluate if the observed changes in maternal dietary habits may affect pregnancy complications and newborn health. Thirdly, data from women enrolled during the COVID-19 pandemic referred to the period between December 2020 and January 2022. By contrast, the majority of previous studies were conducted during or immediately after COVID-19 lockdown. Thus, comparisons between our findings and those from other studies must be made with caution. Finally, dietary data were collected by a FFQ that—although it is a simple, time- and cost-efficient instrument—is prone to measurement errors and inaccuracies [51]. Moreover, dietary data referred to the early phase of pregnancy, not allowing to evaluate changes in maternal dietary habits throughout pregnancy. Previous studies, however, proved that maternal dietary habits may not change largely in this period, despite an increase in total energy intake [52,53]. Regarding PCA, it should be noted that the retained dietary patterns explained only the 15.6% of total variance among food categories. Nonetheless, we used well-established criteria to retain dietary patterns, which were also consistent with those obtained by previous studies [34,54].

## 5. Conclusions

Our findings support the hypothesis that pandemic had a negative impact on maternal diet, showing that the adherence to MD was lower among women enrolled during the COVID-19 pandemic. In particular, maternal dietary habits diverged from a prudent dietary pattern characterized by a higher level of consumption of healthy foods. As such, in order to develop effective public health strategies, further efforts are still needed to deeply explore the main determinants of nutritional behaviors and the long-term effects of COVID-19 on maternal and child health.

## Figures and Tables

**Figure 1 nutrients-14-03380-f001:**
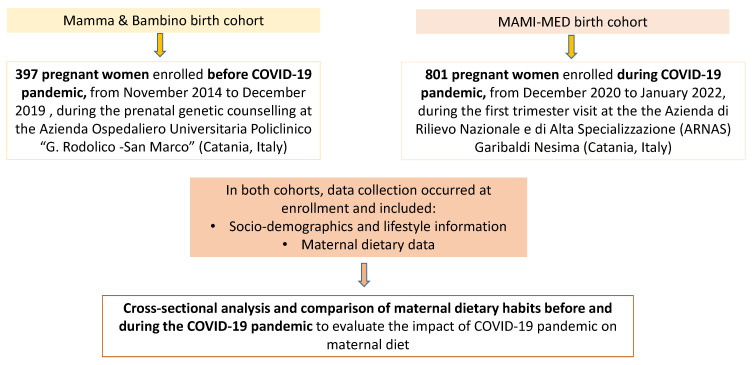
Flow chart of the study design and methodology.

**Figure 2 nutrients-14-03380-f002:**
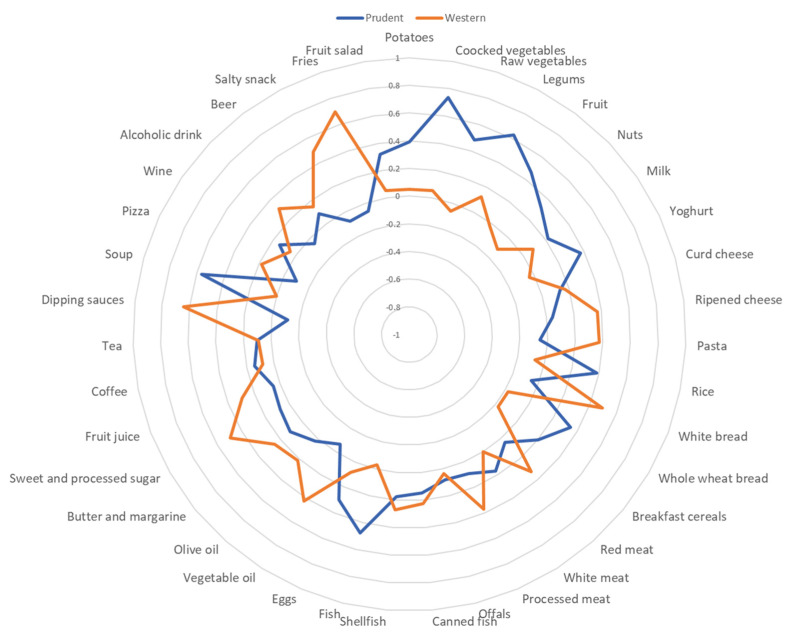
Radar graph of factor loadings for each dietary pattern.

**Table 1 nutrients-14-03380-t001:** Comparison of maternal characteristics before and during the COVID-19 pandemic.

Characteristics	Before COVID-19 Pandemic (*n* = 397)	During COVID-19 Pandemic (*n* = 801)	*p*-Value ^a^
Age, years ^b^	37.0 (4.0)	31.0 (7.0)	<0.001
Gestational age at recruitment ^b^	16.0 (4.0)	12.0 (0)	<0.001
Educational level			
Low educational level ^c^	18.9%	26.8%	0.002
Medium-high educational level	81.1%	73.2%	
Employment status			
Employed	59.2%	50.7%	0.005
Unemployed	40.8%	49.3%	
Smoking (% yes)	43.3%	41.1%	0.460
Pre-pregnancy BMI, kg/m^2 b^	22.8 (5.2)	23.2 (5.9)	0.049
Pre-pregnancy BMI categories			
Underweight	6.6%	5.4%	0.140
Normal weight	64.6%	59.2%	
Overweight	18.2%	22.8%	
Obese	10.6%	12.6%	

^a^ *p*-values are based on the Mann–Whitney test for quantitative variables, or Chi-squared test for categorical variables, ^b^ data are reported as median interquartile range (IQR), ^c^ defined as ≤ 8 years of school. Abbreviations: BMI, body mass index.

**Table 2 nutrients-14-03380-t002:** Comparison of maternal consumption of main food categories before and during COVID-19 pandemic.

Dietary Consumption	Before COVID-19 Pandemic (*n* = 397)	During COVID-19 Pandemic (*n* = 801)	*p*-Value ^a^
Cereals			0.227
Poor	52.4%	48.7%	
Ideal	47.6%	51.3%	
Vegetables			<0.001
Poor	37.5%	56.1%	
Ideal	62.5%	43.9%	
Legumes			<0.001
Poor	42.3%	52.9%	
Ideal	57.7%	47.1%	
Fruits			<0.001
Poor	32.7%	58.6%	
Ideal	67.3%	41.4%	
Fish			<0.001
Poor	41.6%	54.4%	
Ideal	58.4%	45.6%	
Dairy products			<0.001
Poor	45.7%	58.9%	
Ideal	54.3%	41.1%	
Meat			0.527
Poor	50.9%	48.9%	
Ideal	49.1%	51.1%	
Unsaturated/saturated lipids ratio			0.951
Poor	50.1%	49.9%	
Ideal	49.9%	50.1%	

^a^ *p*-values are based on the Chi-Square test.

## Data Availability

The data that support the findings of this study are available from the corresponding author, upon reasonable request.

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
