# Peer review of "The Impact of the COVID-19 Pandemic on Dietary Patterns of Pregnant Women: A Comparison between Two Mother-Child Cohorts in Sicily, Italy"

_nutrients, 2022, doi:10.3390/nu14163380_

Round 1
Reviewer 1 Report
This manuscript describes a comparison of dietary patterns of pregnant women before and after the COVID-19 pandemic. I have some comments to the author.
Comment 1. Results 3.2. The impact of COVID-19 pandemic on maternal consumption of foods mentions maternal dietary consumption before and during COVID-19 pandemic about vegetables, fruits, etc. in the text, but the results such as tables showing this are Isn't there?
Comment 2. Similar to the comments above, are there any tables or other results that show this for finding 3.3. The impact of COVID-19 pandemic on maternal adherence to Mediterranean diet?
Comment 3. The authors' group uses similar analytical methods, as also shown in ref. 25 & ref.26. A past report also shows a “Radar graph of factor loadings for each dietary pattern” and compares “Prudent” and “Western”. Then, the result of this time and
What is the relationship with the pattern in the pre-COVID-19 pandemic papers ref. 25 & ref.26?
Comment 4. This is also related to comment 3 above, but the pattern of "prudent" seems to change slightly in each paper. As described in the text, I understand that these are the results obtained from a cohort obtained from a series of trials, but the difference between each time in the pattern of "Prudent" is (especially before pandemic) what does it mean? What kind of pattern is the “healthy dietary pattern” in the conclusion? Is the “healthy dietary pattern” equal to the "prudent" pattern? If so, wouldn't the prudent pattern be pretty much the same each time?
Comment 5. Even if there were differences in maternal dietary patterns before and after the COVID-19 pandemic, how much did the consequences affect pregnancy complications, birth problems, and fetal health?
Author Response
Dear Editor,
please consider the revised version of the manuscript entitled “The impact of COVID-19 pandemic on dietary patterns of pregnant women: a comparison between two mother-child cohorts in Sicily, Italy” in which we have considered all comments and suggestions from reviewers. We have also reduced the citation rate for the papers of Nutrients journal and the similarity index. This letter is intended for the convenience of the Editor and Reviewers and contains the list of the requested changes. The following list of changes and answers to comments of Reviewers addresses all revisions made in the manuscript (in red font).
Reviewer 1
This manuscript describes a comparison of dietary patterns of pregnant women before and after the COVID-19 pandemic. I have some comments to the author
Answer: We would like to take this opportunity to thank the Reviewer for his/her comments and suggestions which helped us in improving our manuscript.
Comment: Results 3.2. The impact of COVID-19 pandemic on maternal consumption of foods mentions maternal dietary consumption before and during COVID-19 pandemic about vegetables, fruits, etc. in the text, but the results such as tables showing this are Isn't there?
Answer: We are grateful for this comment and, as suggested, we added the Table 2 on the maternal consumption of main food categories before and during COVID-19 pandemic.
Comment: Similar to the comments above, are there any tables or other results that show this for finding 3.3. The impact of COVID-19 pandemic on maternal adherence to Mediterranean diet?
Answer: In the paragraph 3.3, we compared the adherence to Mediterranean diet (MD) before and during the COVID-19 pandemic. All findings are described in the text. In particular, we first compared the adherence to MD in terms of Mediterranean Diet Score (MDS). In the new version of the manuscript we have reported the comparison of those values as median and interquartile range, as well as the output of a linear regression model adjusting for covariates. Next, we compared the proportion of women in each category of adherence to MD, showing the crude OR for the low adherence to MD. Also, in this case, we reported the output of a logistic regression model adjusting for covariates.
Comment: The authors' group uses similar analytical methods, as also shown in ref. 25 & ref.26. A past report also shows a “Radar graph of factor loadings for each dietary pattern” and compares “Prudent” and “Western”. What is the relationship with the pattern in the pre-COVID-19 pandemic papers ref. 25 & ref.26? Moreover, the pattern of "prudent" seems to change slightly in each paper. As described in the text, I understand that these are the results obtained from a cohort obtained from a series of trials, but the difference between each time in the pattern of "Prudent" is (especially before pandemic) what does it mean? What kind of pattern is the “healthy dietary pattern” in the conclusion? Is the “healthy dietary pattern” equal to the "prudent" pattern? If so, wouldn't the prudent pattern be pretty much the same each time?
Answer: As described in the methods of the present work, as well as in a previous study (ref. 26), the Principal Component Analysis is one of the most commonly used exploratory methods to derive a posteriori dietary patterns from a multivariate dataset. This is a data-driven approach that allows to obtain dietary patterns using dietary information of the study population. For this reason, the dietary patterns we derived in the current work are pretty much the same as those obtained previously, though with some differences. However, as suggested by the Reviewer, we discussed this point thoroughly in the discussion section. Please also consider, accordingly, changes in the conclusion section.
Comment: Even if there were differences in maternal dietary patterns before and after the COVID-19 pandemic, how much did the consequences affect pregnancy complications, birth problems, and fetal health?
Answer: Thank you for this comment. Our findings support the hypothesis that COVID-19 had a negative impact on maternal diet, with a lower adherence to MD among women enrolled during the pandemic. Moreover, we found that during the COVID-19 pandemic women diverged from a prudent dietary pattern, which is considered a healthy model. However, as reported in the limitation and conclusion sections, further efforts are needed to deeply investigate the effect of these changes on adverse pregnancy outcomes. In fact, the majority of studies investigated the impact of maternal COVID-19 infection on pregnancy outcomes and there are no studies focused on the impact of maternal diet changes on pregnancy complications, birth problems, and fetal health. Thus, our analysis could be useful to assess the impact of COVID-19 on maternal diet, in order to further investigate the relationship between unhealthy diet and adverse pregnancy outcomes during COVID-19 pandemic. Please consider the new conclusion section: “Our findings support the hypothesis that pandemic had a negative impact on maternal diet, showing that the adherence to MD was lower among women enrolled during the COVID-19 pandemic. In particular, in this case, maternal dietary habits diverge from a prudent dietary pattern characterized by a higher level of consumption of healthy foods. As such, in order to develop effective public health strategies, further efforts are still needed to deeply explore the main determinants of nutritional behaviours and the long-term effects of COVID-19 on maternal and child health.”
Reviewer 2 Report
Thank you very much for allowing me to review the article “The impact of COVID-19 pandemic on dietary patterns of pregnant women: a comparison between two mother-child cohorts in Sicily, Italy.” (nutrients-1846406).
This study is presented to “Nutrients” for the section “Nutrition and Public Health” for the Special Issue “The association of Dietary Factors and Disease Risk”.
The context in which this study is carried out is of great interest. The diet during pregnancy is essential for the health of the mother and the newborn, however during the COVID-19 pandemic, modifications of different meaning were identified in relation to the reference diet.
The aim of this article is to evaluate the impact of COVID-19 on the diet of pregnant women, using data from two mother-child cohorts in Sicily (Italy).
They identify how the confinement due to the COVID-19 pandemic modified the diet during pregnancy and the elements that were modified from said diet in Catania, Sicily, Italy.
The introduction is clear and well thought out and informs about the elements that justify the study.
The objective is clear and well defined, it´s also corresponds to the title of the work, but it´s no clear with the methodology
Material and methods
They used two cohorts: “Mamma & Bambino” and the “MAMI-MED” cohorts, during the period 2014-2019. Also they used “MAMI-MED” cohort during the first trimester pregnant visit (4 to 20 weeks of gestation).
I suggest that the number of participating women be reported for each of the cohorts that are combined in this study and whether they correspond to the same women or are different. The dietary surveys are only carried out once, please clarify this methodological aspect.
The design should be clarified because, although it is based on a follow-up cut, they use a cross-sectional design before and after the pandemic, analyzing it as a control case nested on said cohorts. Please clarify this aspect of the design as it creates a lot of confusion about the results.
I suggest they focus on women since you don't really look at any data on children.
I suggest that a flow chart be presented to make the methodology used more understandable.
Results
The differences in the fundamental characteristics of pregnant women before and after the pandemic inform us that they are different groups and therefore the results must be adjusted for these variables since they can be effect modifiers.
I suggest presenting a table with the dietary differences identified from both the crude ORs and the adjusted ORs. This is the aim of this study.
Discussion
It is a very interesting discussion, but we still have the design problem to be able to interpret the results, I think it should be taken into account in a very important way.
Author Response
Dear Editor,
please consider the revised version of the manuscript entitled “The impact of COVID-19 pandemic on dietary patterns of pregnant women: a comparison between two mother-child cohorts in Sicily, Italy” in which we have considered all comments and suggestions from reviewers. We have also reduced the citation rate for the papers of Nutrients journal and the similarity index. This letter is intended for the convenience of the Editor and Reviewers and contains the list of the requested changes. The following list of changes and answers to comments of Reviewers addresses all revisions made in the manuscript (in red font).
Reviewer 2
Comment: Thank you very much for allowing me to review the article “The impact of COVID-19 pandemic on dietary patterns of pregnant women: a comparison between two mother-child cohorts in Sicily, Italy.” (nutrients-1846406).
This study is presented to “Nutrients” for the section “Nutrition and Public Health” for the Special Issue “The association of Dietary Factors and Disease Risk”. The context in which this study is carried out is of great interest. The diet during pregnancy is essential for the health of the mother and the newborn, however during the COVID-19 pandemic, modifications of different meaning were identified in relation to the reference diet. The aim of this article is to evaluate the impact of COVID-19 on the diet of pregnant women, using data from two mother-child cohorts in Sicily (Italy). They identify how the confinement due to the COVID-19 pandemic modified the diet during pregnancy and the elements that were modified from said diet in Catania, Sicily, Italy. The introduction is clear and well thought out and informs about the elements that justify the study.
Answer: We would like to take this opportunity to thank the Reviewer for his/her comments and suggestions which helped us in improving our manuscript.
Comment: The objective is clear and well defined, it´s also corresponds to the title of the work, but it´s no clear with the methodology. They used two cohorts: “Mamma & Bambino” and the “MAMI-MED” cohorts, during the period 2014-2019. Also they used “MAMI-MED” cohort during the first trimester pregnant visit (4 to 20 weeks of gestation). I suggest that the number of participating women be reported for each of the cohorts that are combined in this study and whether they correspond to the same women or are different. The dietary surveys are only carried out once, please clarify this methodological aspect.
Answer: We apologize if the methodology was not clearly written. We used data from two cohorts, the “Mamma & Bambino” and the “MAMI-MED”. In particular, we included pregnant women enrolled before the COVID-19 pandemic (2014-2019) in the “Mamma & Bambino” cohort, and different pregnant women enrolled during the COVID-19 pandemic (2020-2022) in the “MAMI-MED” cohort. Then, we compared data from the two cohorts in order to evaluate the impact of COVID-19 pandemic on maternal diet. To do this, in each cohort, socio-demographics and lifestyles information was assessed at recruitment by trained epidemiologists through structured questionnaires. Moreover, in each cohort, maternal diet was assessed at recruitment using a Food Frequency Questionnaire. Please consider all changes made in the 2.1 and 2.2 paragraphs.
Comment: The design should be clarified because, although it is based on a follow-up cut, they use a cross-sectional design before and after the pandemic, analyzing it as a control case nested on said cohorts. Please clarify this aspect of the design as it creates a lot of confusion about the results.
Answer: We apologize if the study design was not clearly described. As suggested, we better specified in the methods section that the present study is a cross-sectional analysis, in which we used data from two birth cohorts (https://www.birthcohorts.net/birthcohorts/birthcohort/?id=243). In particular, we compared maternal diet before and during the COVID-19 pandemic, using data from two different birth cohorts – the “Mamma & Bambino” and the “MAMI-MED” – which enrolled pregnant women respectively before and during the COVID-19 pandemic.
Comment: I suggest they focus on women since you don't really look at any data on children.
Answer: As suggested by the title, we focused on the impact of COVID-19 pandemic on maternal diet using data from two birth cohorts. Further analysis will be done to evaluate the effects of maternal diet changes on pregnancy outcomes and newborns health.
Comment: I suggest that a flow chart be presented to make the methodology used more understandable.
Answer: As suggested, we have provided a flow chart (Figure 1) on methodology used in our analysis.
Comment: Results. The differences in the fundamental characteristics of pregnant women before and after the pandemic inform us that they are different groups and therefore the results must be adjusted for these variables since they can be effect modifiers.
Answer: As reported in the results, the present cross-sectional study included 1198 pregnant women, 397 out of which recruited in the “Mamma & Bambino” cohort before the beginning of the COVID-19 pandemic (median age = 37 years). The remaining 801 pregnant women, conversely, were enrolled in the “MAMI-MED” cohort, during the COVID-19 pandemic (median age = 31 years). The comparison of maternal characteristics before and after the beginning of COVID-19 pandemic suggested differences in terms of age (p<0.001), gestational age at recruitment (p<0.001), pre-pregnancy BMI (p=0.049), educational level (p=0.002) and employment status (p=0.005). Thus, we added a sentence in which we specified that all analyses were adjusted for these confounders. Moreover, we specified it in all the paragraphs of results.
Comment: I suggest presenting a table with the dietary differences identified from both the crude ORs and the adjusted ORs. This is the aim of this study.
Answer: As suggested, we added the Table 2 showing dietary differences between women enrolled before and during the COVID-19 pandemic. In the main text, we also reported crude and adjusted ORs for all the analyses performed in this work.
Comment: Discussion. It is a very interesting discussion, but we still have the design problem to be able to interpret the results, I think it should be taken into account in a very important way.
Answer: Thank you for the comment. As described in discussion, our work had also some limitations that should be considered when interpreting our results. Firstly, our study was conducted comparing dietary habits from two different cohorts. Although we performed our analyses adjusting for potential confounders, the presence of residual confounders cannot be completely ruled out. Moreover, the limited sample size did not allow us to perform additional analyses. Thus, further analyses are needed to corroborate our findings.
Round 2
Reviewer 1 Report
Thank you for correcting the paper.
Reviewer 2 Report
After reviewing the new version of the manuscript article “The impact of COVID-19 pandemic on dietary patterns of pregnant women: a comparison between two mother-child cohorts in Sicily, Italy.” (nutrients-1846406), incorporating the reviewers' suggestions and the authors' responses, I find the article to be comprehensive and informative about the change in diet of pregnant women due to the COVID-19 pandemic.